# Benzalkonium Chloride, Even at Low Concentrations, Deteriorates Intracellular Metabolic Capacity in Human Conjunctival Fibroblasts

**DOI:** 10.3390/biomedicines10092315

**Published:** 2022-09-18

**Authors:** Yuri Tsugeno, Tatsuya Sato, Megumi Watanabe, Masato Furuhashi, Araya Umetsu, Yosuke Ida, Fumihito Hikage, Hiroshi Ohguro

**Affiliations:** 1Departments of Ophthalmology, School of Medicine, Sapporo Medical University, Sapporo 060-8556, Japan; 2Departments of Cardiovascular, Renal and Metabolic Medicine, Sapporo Medical University, Sapporo 060-8556, Japan; 3Departments of Cellular Physiology and Signal Transduction, Sapporo Medical University, Sapporo 060-8556, Japan

**Keywords:** 3D spheroid cultures, human conjunctival fibroblast (HconF), benzalkonium chloride (BAC)

## Abstract

The objective of this study was to clarify the effects of benzalkonium chloride (BAC) on two-dimensional (2D) and three-dimensional (3D) cultures of human conjunctival fibroblast (HconF) cells, which are in vitro models replicating the epithelial barrier and the stromal supportive functions of the human conjunctiva. The cultured HconF cells were subjected to the following analyses in the absence and presence of 10^−5^% or 10^−4^% concentrations of BAC; (1) the barrier function of the 2D HconF monolayers, as determined by trans-endothelial electrical resistance (TEER) and FITC dextran permeability, (2) real-time metabolic analysis using an extracellular Seahorse flux analyzer, (3) the size and stiffness of 3D HconF spheroids, and (4) the mRNA expression of genes that encode for extracellular matrix (ECM) molecules including collagen (COL)1, 4 and 6, and fibronectin (FN), α-smooth muscle actin (α-SMA), ER stress related genes including the X-box binding protein-1 (XBP1), the spliced XBP1 (sXBP1) glucose regulator protein (GRP)78, GRP94, and the CCAAT/enhancer-binding protein homologous protein (CHOP), hypoxia inducible factor 1α (HIF1α), and Peroxisome proliferator-activated receptor gamma coactivator 1α (PGC1α). In the presence of BAC, even at low concentrations at 10^−5^% or 10^−4^%, the maximal respiratory capacity, mitochondrial respiratory reserve, and glycolytic reserve of HconF cells were significantly decreased, although the barrier functions of 2D HconF monolayers, the physical properties of the 3D HconF spheroids, and the mRNA expression of the corresponding genes were not affected. The findings reported herein highlight the fact that BAC, even such low concentrations, may induce unfavorable adverse effects on the cellular metabolic capacity of the human conjunctiva.

## 1. Introduction

It is well known that medical instillation therapy in the treatment of chronic ocular diseases including glaucoma, dry eye, and others can cause ocular surface adverse effects (OSAE) that can affect the eyelids, conjunctiva, and/or corneal epithelium [1,2,3,4], with the following symptoms often being associated with these conditions; redness, irritation, burning, fatigue, deteriorating visual acuity, infections, and others. As a possible causative factor, it has been suggested that for OSAE, toxicity by benzalkonium chloride (BAC), which is most commonly used in concentrations of 4~2 × 10^−2^% as a preservative in topical ophthalmic formulations [2,3,4,5,6], toward ocular tissue cells is the likely cause. The threshold concentration for inducing such toxic effects has been estimated to be approximately ~5 × 10^−3^% based upon several in vitro and in vivo studies showing that BAC exposure induced (1) a reduced survival of corneal cells [7,8,9,10,11,12], conjunctival cells [7,8,13,14], trabecular meshwork (TM) cells [15,16], and ciliary epithelial cells [9,13,15,17], loss of conjunctival goblet cells [13,14], (2) delayed corneal epithelial wound healing [18], (3) the induction of lymphocyte infiltration into conjunctival tissue [13,17], and (4) an increase in the levels of inflammatory cytokines in ocular tissues [9,10,12]. Alternatively, several clinical studies have reported that these BAC-induced adverse effects may partially be reversible upon withdrawal of the exposure to BAC [19,20,21,22,23,24].

However, in contrast, another previous study indicated that a short time exposure (30 min) to much lower concentrations of BAC ranging from 5 × 10^−5^% ~ or 10^−3^% toward immortalized human corneal epithelial cells (HCEs) caused DNA double-strand breaks (DSBs) and these breaks were concentration-dependent [25]. This information rationally suggests that BAC-induced risk may be evoked, even when much lower concentrations (5 × 10^−5^% ~ or 10^−3^%) of BAC are used as compared with the above estimated threshold levels. Therefore, to study the cytotoxic effects of BAC at much lower concentrations (10^−5^% and 10^−4^%) toward conjunctival tissues, we employed recently established in vitro models for the epithelial barrier and the stromal supportive functions of the human conjunctiva using two-dimension (2D) and three-dimension (3D) spheroid cultures of the human conjunctival fibroblasts (HconF) cells [26], and the following analyses were carried out: (1) barrier functions of 2D cultured HconF monolayers by trans-endothelial electron resistance (TEER) and FITC dextran permeability measurements, (2) measurements of real-time mitochondrial and glycolytic cellular function, (3) measurements of the size and hardness of the 3D HconF spheroids, and (4) quantitative PCR of major extracellular matrix (ECM) molecules, including collagen (COL)1, 4 and 6, and fibronectin, α-smooth muscle actin (α-SMA), ER stress related genes including the X-box binding protein-1 (XBP1), the spliced XBP1 (sXBP1) glucose regulator protein (GRP)78, GRP94, and the CCAAT/enhancer-binding protein homologous protein (CHOP), hypoxia inducible factor 1α (HIF1α), and Peroxisome proliferator-activated receptor gamma coactivator 1 (PGC1α). 

## 2. Materials and Methods

### 2.1. 2D Cell and 3D Spheroid Cultures of Human Conjunctival Fibroblasts (HconF) 

BAC powder (CAS#8001-54-5, Nacalai Tesque, Kyoto, Japan) was dissolved in DMEM (FUJIFILM Wako Pure Chemical Corporation, Osaka, Japan) adjusted to a concentration of 0.2%, and further diluted in 10% FBS (BioWest, Nuaillé, France). As a negative control, the same diluent was used. In the presence 10^−5^% or 10^−4^% BAC, or only diluent, HconF cells (ScienCell Research laboratories, CA U.S.A.) were 2D cultured and further maintained or subjected to 3D spheroid culturing for 6 days using hanging drop culture plates (# HDP1385, Sigma-Aldrich, St Louis, MO, USA), as described in our previous study [26]. Briefly, 2D cultured HconF cells in 150 mm 2D culture dishes at 37 °C in the Fibroblast Medium (FM, Cat. #2301, ScienCell Research laboratories, Carlsbad, CA, USA) [26] were maintained by changing the medium every other day. Alternatively, these 2D cultured HconF cells were washed with phosphate buffered saline (PBS), detached using 0.05% Trypsin/EDTA (FUJIFILM Wako Pure Chemical Corporation, Osaka, Japan), and re-suspended in the Fibroblast Medium supplemented with 0.25% methylcellulose (Methocel® A4M, Sigma-Aldrich, St Louis, MO, USA). Then, approximately 20,000 HconF cells in 28 μL of the suspension were subjected to each well of the 3D hanging drop culture plate (# HDP1385, Sigma-Aldrich) (Day 0). Thereafter, on each following day until Day 6, half of the medium (14 μL) was exchanged by fresh medium.

### 2.2. Analysis of the Barrier Function of 2D HconF Cell Monolayers by TEER and FITC Dextran Permeability 

In the absence and presence of 10^−5^% or 10^−4^% BAC, HconF cells were 2D cultured using a TEER plate (0.4 μm pore size and 12 mm diameter; Corning Transwell, Sigma-Aldrich) at 37 °C in the Fibroblast Medium as above. On Day 6, the TEER values between the 2D HconF monolayer were measured using an electrical resistance system (KANTO CHEMICAL CO. INC., Tokyo, Japan), and FITC-dextran permeability was estimated by measuring the fluorescence intensity that had permeated through the membrane from the basal compartment to the apical compartment during a period of 60 min as described in our previous study [27].

### 2.3. Seahorse Real-Time Bio-Cellular Metabolic Function Analysis of the 2D HconF Cells

As the bio-cellular function of 2D HconF cells, their oxygen consumption rate (OCR) and the extracellular acidification (ECAR) of 10^−5^% or 10^−4^% BAC-treated or un-treated 2D HconF cells were evaluated by a Seahorse XFe96 Bioanalyzer (Agilent Technologies, Santa Clara, CA, USA) as described in the previous studies [28,29]. In brief, approximately 20,000 2D cultured HconF cells per well were set in an XFe96 Cell Culture Microplate (Agilent Technologies, #103794-100). The plate was centrifuged at 1600× *g* for 10 min, and the culture medium was replaced with 180 µL of assay buffer (Seahorse XF DMEM assay medium with 5.5 mM glucose, 2.0 mM glutamine, 1.0 mM sodium pyruvate (pH 7.4, Agilent Technologies, #103575-100)). Then, the assay plate was incubated in a CO_2_-free incubator at 37 °C for 1 h prior to the assay. OCR and ECAR were simultaneously measured on a Seahorse XFe96 Bioanalyzer under a 3-min-mixing and 3-min-measuring protocols using the following sequential injection of oligomycin (final concentration: 2.0 μM), carbonyl cyanide p-trifluoromethoxyphenylhydrazone (FCCP, final concentration: 5.0 μM), rote-none/antimycin A mixture (final concentration: 1.0 μM), and 2-deoxyglucose (2-DG, final concentration: 10 mM). Valses of OCR and ECAR were normalized to the total protein per well after completion of assay.

### 2.4. Evaluation of the Size and Hardness of HconF Cell 3D Spheroids

The analyses of the physical properties, size, and hardness of the HconF 3D spheroids were performed as reported in our previous studies [30,31]. Briefly, the mean sizes of the 3D spheroids were measured using an inverted microscope (Nikon ECLIPSE TS2; Tokyo, Japan). Alternatively, for the hardness measurement, a single living 3D spheroid was placed on a 3-mm × 3-mm plate and compressed to achieve a 50% deformation during 20 seconds using a micro-compressor (MicroSquisher, CellScale, Waterloo, ON, Canada). The required force (μN) was measured, and force/displacement (μN/μm) was calculated.

### 2.5. Other Analytical Methods

Total RNA was extracted from the 2D or 3D cultured HconF cells and those were subjected to reverse transcription and real-time PCR as previously reported [32,33] using specific primers and probes (Appendix A).

As described in a recent report [32,33], all statistical analyses were performed using Graph Pad Prism 8 (GraphPad Software, San Diego, CA, USA). A significant difference at less than 0.05 between experimental groups by ANOVA followed by a Tukey’s multiple comparison test was determined as statistically significant.

## 3. Results

Among ocular surfaces, the conjunctiva is well known to be involved in two different biological roles, namely, (1) a biological barrier by conjunctival epithelium [34] and (2) ocular tissue support, repair, and remodeling by conjunctival stroma [35,36]. In the current study, to investigate the effects of low concentrations of BAC at 10^−5^% or 10^−4^% on these conjunctival functions, under which no significant cytotoxicity was observed (Appendix A), we employed our recently established in vitro model using 2D and 3D cultures of HconF cells [26] which are thought to be related to the epithelial and stromal functions, respectively. 

In a previous study we showed that, as the barrier function of the corneal epithelium, TEER values had significantly deteriorated even after a 20 min exposure to 10^−3^% of BAC [11]. Furthermore, 10^−4^% BAC also caused the significant up-regulation of IL-6 and IL-8 genes of 2D and the Matrigel^®^-assisted 3D spheroids of human trabecular meshwork (HTM) cells [37], although cytotoxic effects by BAC (2 × 10^−2^%) were not detected in the 3D corneal epithelial culture model [38]. However, in our established in vitro 3D spheroid model using HconF cells [26], HTM cells [31,39], human corneal stromal fibroblasts (HCSFs) [40] and human orbital fibroblast (HOFs) [41,42], several drugs including those treated with PGF2α, ROCK inhibitors, and others failed to induce cytotoxic effects but their physical properties were greatly modulated. These collective findings suggest that our established measurements of these physical properties of the 3D spheroids should be more sensitive in estimating cellular biological aspects, and therefore, it would be possible to use this methodology to evaluate effects of concentrations of BAC as low as 10^−5^% or 10^−4^%. Thus, initially, to study such low concentrations of BAC-induced effects on barrier function, TEER and FITC dextran permeability measurements of 2D HconF cell monolayers were conducted. As shown in Figure 1, both measurements were not affected at all by the presence of both 10^−5^% and 10^−4^% of BAC. Next, to estimate the influences of these low concentrations of BAC toward the tissue supportive functions of conjunctiva, the 3D HconF spheroid model was used. The result indicated that their physical properties, size, and stiffness were also not altered by these concentrations of BAC as similarly to the barrier functions of the 2D monolayer as above (Figure 2).

To elucidate additional aspects related to the BAC-induced effects on the cellular physiology of the 2D HconF cells, mitochondria- and glycolysis-related functions were evaluated by a Seahorse real-time bio-cellular analyzer, since previous studies showed that upon BAC exposure, the OCR of rat liver mitochondria was substantially deteriorated [43], and glycolysis in bacteria was greatly modulated [44]. However, mitochondrial maximal respiration, mitochondrial respiratory reserve, and glycolytic reserve in HconF cells were significantly decreased by the presence of BAC, even at low concentrations, suggesting that BAC can adversely affect intracellular metabolic capacity (Figure 3A–D). The energy map visually indicated that metabolic response induced by FCCP-induced stress in HconF cells was impaired even with 10^−4^% BAC exposure, while BAC had no effect on baseline metabolism at either 10^−5^% and 10^−4^% concentrations (Figure 3E). These functional assays suggested that cellular mitochondrial and glycolytic functions were already affected even though the barrier functions by TEER and FITC dextran permeability of the 2D HconF monolayer and physical properties of the 3D HconF spheroid were not influenced in the presence of 10^−5^% and 10^−4^% BAC.

To study this issue further, the expressions of several genes including major ECM molecules including COL 1,4 and 6, FN and α-SMA, ER stress related factors, HIF1α, and PGC1α were analyzed by qPCR. As shown in Figure 4, Figure 5 and Figure 6, mRNA expression of these molecules was not altered by these concentrations of BAC. Taken together with these gene expression analyses, Seahorse real time bio-metabolic measurements related to mitochondrial and glycolysis may be much more sensitive for evaluation of the physiological states of the living cells as compared with gene expressions of several related factors as well as physical analyses. Thus, these collective results suggest that some functional and morphological abnormality may be induced within human conjunctiva, even when such low concentrations of BAC are being used, especially in case with their long-term exposure.

## 4. Discussion

Since BAC, a polyquaternary ammonium detergent, was registered as a safe additive by the Environmental Protection Agency (EPA) in the United States in 1947 due to its broad-spectrum antimicrobial properties, BAC is now widely used in numerous agricultural, industrial, and clinical products [45,46,47]. Alternatively, it has been pointed out that toxicities could be induced in humans and other animals by oral uptake, by inhalation, or via an epidermal (including the eye) route [48]. Although BAC has not been reported to be carcinogenic, mutagenic, or genotoxic, an in vitro study demonstrated that the BAC concentrations should be controlled to as low as 1 mg/L to avoid possible risks for BAC induced genotoxic effects within plant and mammalian cells [49]. Furthermore, considerable cell toxicity toward human ocular and intranasal cells were observed in vitro on exposure to BAC concentrations as low as 10^−3^% [50] and 45 × 10^−4^% [51] respectively. In fact, the European Chemical Agency (ECHA) labels BAC as “causing severe skin burns and eye damage, is very toxic to aquatic life, is harmful if swallowed, and is harmful in contact with skin”. Therefore, extensive in vivo as well as in vitro studies must be conducted in the case of using drugs containing BAC, including instillations.

Biochemically, it is known that BAC is capable of lysing cell membranes [52] and thus break cell–cell junctions within the corneal epithelium resulting in facilitating the penetration of the topically applied drugs into the anterior chamber [52,53]. It has been reported that BAC in topical ophthalmic formulations can penetrate through the ocular surface into the anterior chamber as well as the optic nerve [54,55]. Alternatively, it is also well known that BAC induces a number of adverse effects by stimulating inflammation in ocular surface tissues such as the conjunctiva [56,57,58,59]. In fact, a recent study reported that BAC exerts dose-dependent BAC toxic effects toward ocular surface epithelial cells using a mouse dry eye model [60]. Among several analyses using corneo-limbal epithelial cells (CLECs) in this study, a 24 h exposure of 10^−2^% BAC resulted in a significantly increased cytotoxicity, as evidenced by LDH assays as well as flow cytometry data on AnnV/PI stained cells as compared with non-treated controls. More interestingly, a significant reduction in both the colony forming efficiency and the colony size of the CLEC cultures were observed when the exposure involved a 10^−4^% BAC solution. Such BAC-induced effects at lower concentrations were also detected by the mitochondrial analysis of human corneal epithelial cells [61], that is, BAC inhibited ATP production (IC50, 5.3 μM; 19 × 10^−5^%) and O_2_ consumption (IC50, 10.9 μM; 39 × 10^−4^%) and this inhibition was concentration-dependent. Taking the cationic property of BAC [52], a quaternary ammonium, into account, this observation is rationally supported by the fact that one of the major targets of BAC in the cell may be the only intracellular negatively charged mitochondria despite little information related to the influence of BAC on mitochondria.

Indeed, the findings presented in this study also demonstrated that significant reductions in mitochondrial and glycolytic reserve capacity are induced on exposure to BAC at concentrations of both 10^−5^% and 10^−4^% in 2D cultured HconF cells despite the fact that other analyses, including the barrier function of 2D monolayers, the physical properties of the 3D spheroids, and the mRNA expressions of several gene of ECM proteins, TIMPs, MMPs, and ER stress related factors were affected. Mitochondrial and glycolytic reserve refers to the ability of a cell to meet increased energy demands, and the retention of plasticity in these metabolic capacities has been reported to prevent cells from being driven into cellular senescence or cell death [62]. Therefore, the reduction of intracellular metabolic reserve induced by BAC exposure may potentially lead to an exacerbated cell dysfunction and cell death in HconF cells. However, as study limitations in the current investigation, additional information such as oxidative stress measurements, for instance, ROS production, or apoptosis will be required to confirm this speculation.

In conclusion, BAC, even at low concentrations, causes the deterioration of intracellular metabolic capacity in HconF cells. Since mitochondria play an indispensable role to maintain proper cellular functions, the effects of BAC noted herein raise great concerns for clinicians who are taking care of patients that are being administered formulations containing BAC, especially in cases of extended use.

## Figures and Tables

**Figure 1 biomedicines-10-02315-f001:**
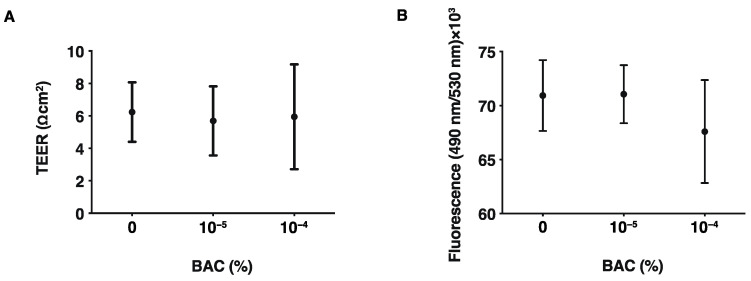
Effects of benzalkonium chloride (BAC) on barrier functions of HconF 2D monolayers. Barrier function based on TEER and FITC dextran permeability measurements were made on HconF cell 2D monolayers in the absence or presence of 10^−5^% or 10^−4^% BAC. Plots of the electric resistance (Ωcm^2^) by TEER and the absorbance of the amounts of permeated fluorescein are shown in (**A**) and (**B**), respectively. Experiments were repeated in triplicate (n = 5 each). All data are expressed; the mean ± the standard error of the mean (SEM). Statistical significance was evaluated by ANOVA followed by a Tukey’s multiple comparison test.

**Figure 2 biomedicines-10-02315-f002:**
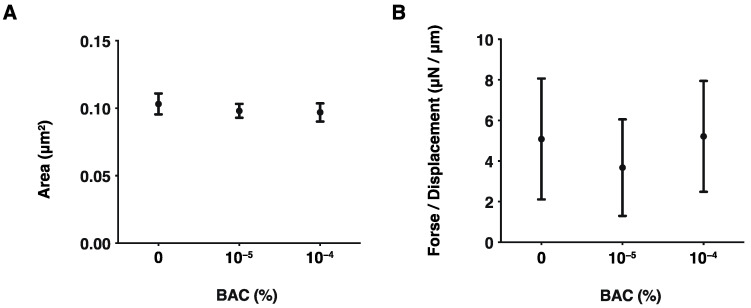
Effects of BAC on the sizes and hardness of the HconF 3D spheroids. Analysis of the size and hardness of 3D HconF spheroids in the absence or presence of 10^−5^% or 10^−4^% BAC. The mean sizes (μm) and the force required to compress a single spheroid to the semidiameter (μN/μm) within 20 seconds are plotted in (**A**) and (**B**), respectively. Experiments were repeated in triplicate using fresh preparations (n = 16 spheroids each). All data are expressed; the mean ± the standard error of the mean (SEM). Statistical significance was evaluated by ANOVA followed by a Tukey’s multiple comparison test.

**Figure 3 biomedicines-10-02315-f003:**
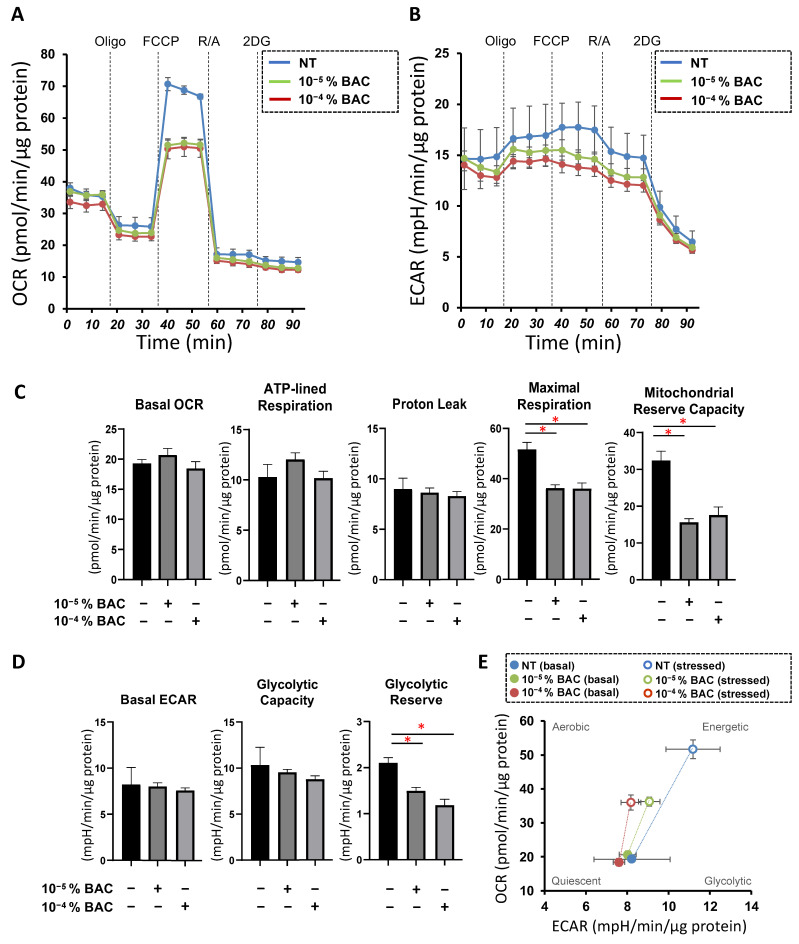
Effects of BAC on the mitochondrial and glycolytic functions in HconF 2D cells. A Seahorse real-time metabolic function analysis of HconF cells in the absence (blue dots, NT) or presence of BAC at concentrations of 10^−5^% (green dots) or 10^−4^% (brown dots) BAC. OCR (**A**) and ECAR (**B**) were measured at baseline and those with injections of sequential supplementation with a complex V inhibitor, oligomycin (Oligo), a protonphore, Carbonyl cyanide-p-trifluoromethoxyphenylhydrazone (FCCP), complex I/III inhibitors, rotenone/antimycin A (R/A), and a hexokinase inhibitor, 2-deoxyglucose (2DG). (**C**) Indicates the parameters of mitochondrial function. Basal OCR was calculated as the difference in OCR at the baseline and after the addition of R/A. ATP-linked respiration was calculated as the difference in OCR at the baseline and after the addition of Oligo. Proton leak was calculated as the difference between OCR after the addition of Oligo and OCR after the addition of R/A. Maximal respiration was calculated as the difference between OCR after the addition of FCCP and after the addition of R/A. Mitochondrial reserve capacity was calculated as the difference in OCR at baseline and after the addition of FCCP. (**D**) Indicates parameters of glycolytic function. Basal ECAR was calculated as the difference in ECAR at baseline and after the addition of 2DG. Glycolytic capacity was calculated as the difference between ECAR after the addition of Oligo and ECAR after the addition of 2DG. Glycolytic reserve was calculated as the difference in ECAR at baseline and after the addition of Oligo. (**E**) Energy map for cells in the absence or presence of 10^−5^% and 10^−4^% BAC. Experiments were performed using fresh preparations (n = 3). Data are expressed; the mean ± the standard error of the mean (SEM). * *p* < 0.05; ANOVA followed by a Tukey’s multiple comparison test.

**Figure 4 biomedicines-10-02315-f004:**
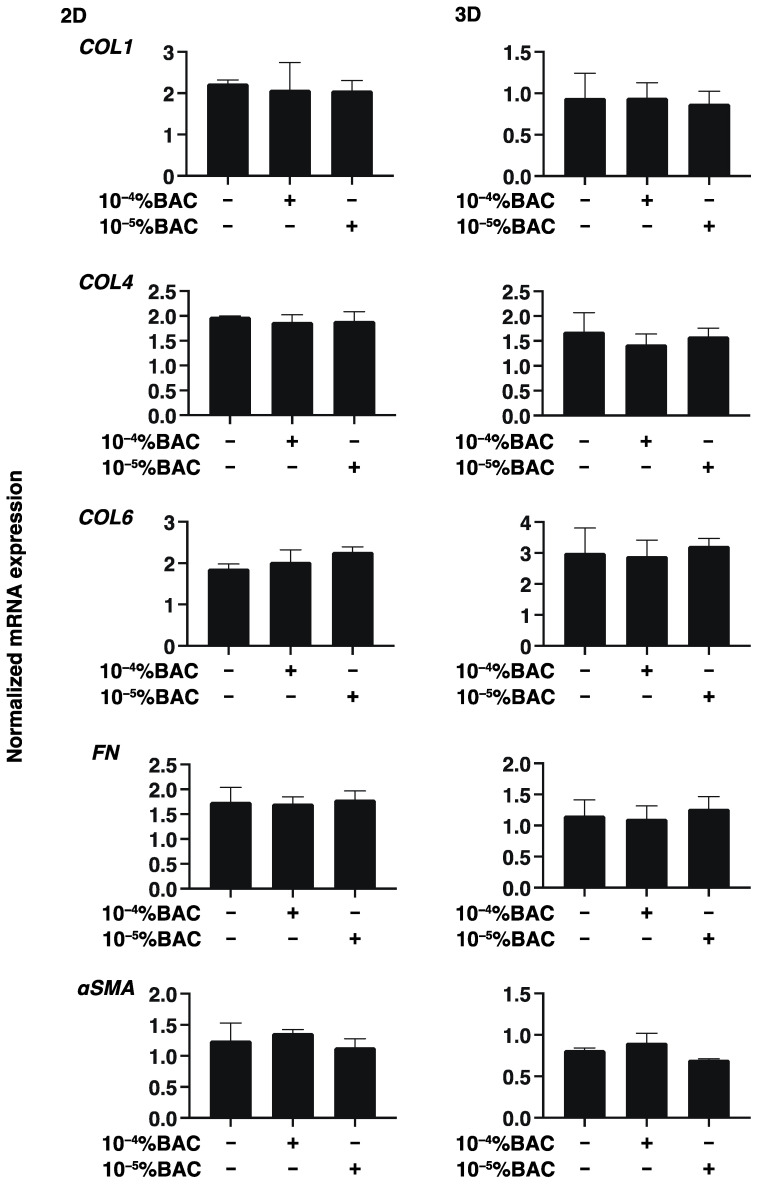
Effects of BAC on the gene expression of ECMs of HconF cells. 2D and 3D HconF cells were subjected to qPCR analysis for ECMs including COL1, COL4, COL6, FN, and aSMA in the absence or presence of 10^−5^% or 10^−4^% BAC. Experiments were repeated in triplicate using 3 different confluent 6-well dishes (2D) or 15 freshly prepared 3D HconF spheroids (3D) in each experimental condition. All data are expressed; the mean ± the standard error of the mean (SEM). Statistical significance was evaluated by ANOVA followed by a Tukey’s multiple comparison test.

**Figure 5 biomedicines-10-02315-f005:**
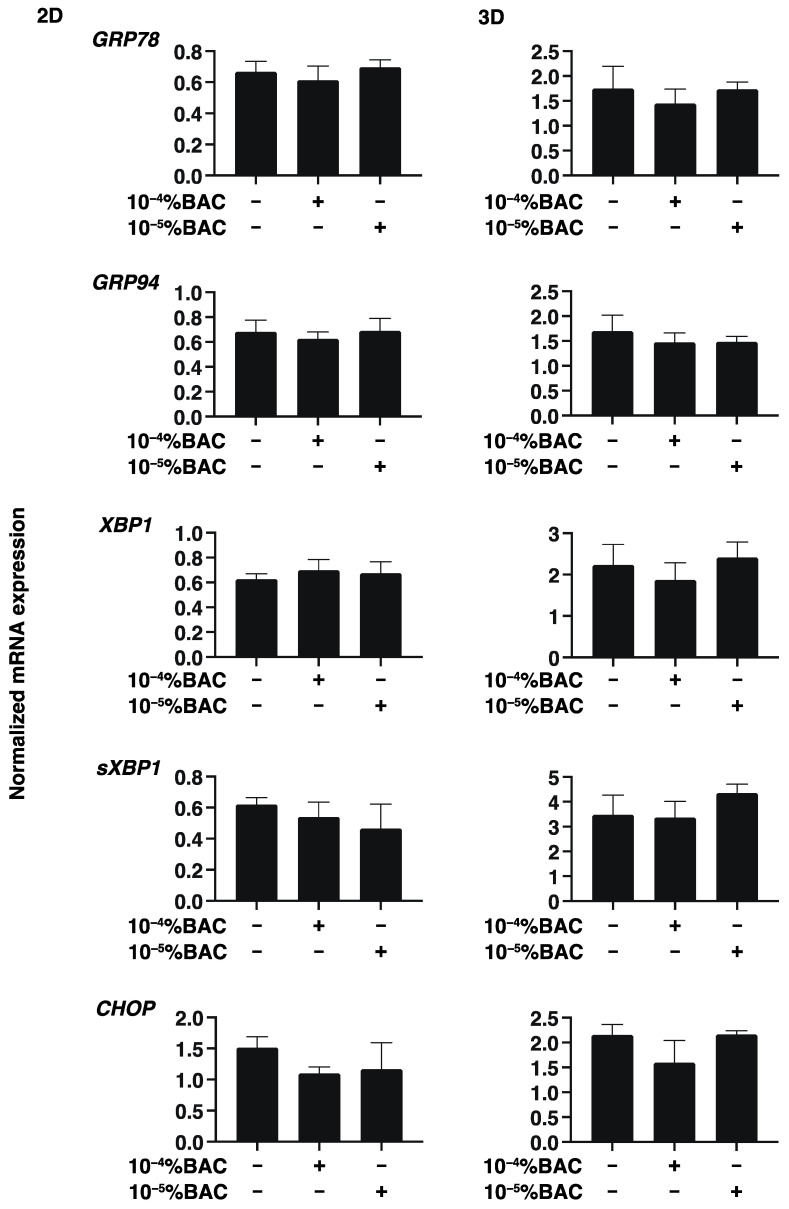
Effects of BAC on the gene expression of ER stress related factors of HconF cells. 2D and 3D HconF cells were subjected to qPCR analysis of ER stress-related genes in the absence or presence of 10^−5^% or 10^−4^% BAC; the glucose regulator protein (GRP)78, GRP94, the X-box binding protein-1 (XBP1), spliced XBP1 (sXBP1), and CCAAT/enhancer-binding protein homologous protein (CHOP). Experiments were repeated in triplicate using 3 different confluent 6-well dishes (2D) or 15 freshly prepared 3D HconF spheroids (3D) in each experimental condition. All data are expressed; the mean ± the standard error of the mean (SEM). Statistical significance was evaluated by ANOVA followed by a Tukey’s multiple comparison test.

**Figure 6 biomedicines-10-02315-f006:**
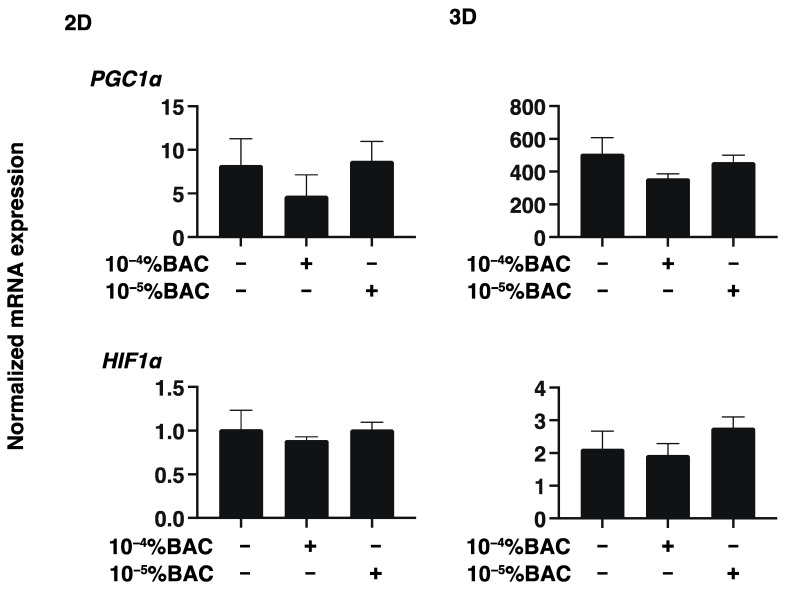
Effects of BAC on the gene expression of HIF1 and PGC1α of HconF cells. 2D and 3D HconF cells were subjected to qPCR analysis in HIF1α and PGC1α and 14 in the absence or presence of 10^−5^% or 10^−4^% BAC. Experiments were repeated in triplicate using 3 different confluent 6-well dishes (2D) or 15 freshly prepared 3D HconF spheroids (3D) in each experimental condition. All data are expressed; the mean ± the standard error of the mean (SEM). Statistical significance was evaluated by ANOVA followed by a Tukey’s multiple comparison test.

## Data Availability

The data that support the findings of this study are available from the corresponding author upon reasonable request.

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
