# Peer review of "Benzalkonium Chloride, Even at Low Concentrations, Deteriorates Intracellular Metabolic Capacity in Human Conjunctival Fibroblasts"

_biomedicines, 2022, doi:10.3390/biomedicines10092315_

Round 1

Reviewer 1 Report

In the presented research, the authors evaluated the effects of benzalkonium chloride on the human conjunctival fibroblast in 2D and 3D. BAC has been shown to cause negative effects in eye cells and yet, is still used as a preservative in some pharmacological formulations intended to be used in ocular diseases. 

The research is sound and straightforward. What is missing, in my opinion is clear identification of the analyzed chemical by its CAS number, and there is not enough data on how the 0.0001 and 0.00001% concentrations of the compound were prepared (what was used as a diluent and therefore what was used as a negative control?). The lack of this basic information prevents this article from being used in any systemic review that may be conducted for this compound. What is more, in the Discussion section, I would like to see the effects of BAC in vivo being better described. ECHA labeled this compound as "causing severe skin burns and eye damage, is very toxic to aquatic life, is harmful if swallowed, and is harmful in contact with skin" therefore, in vivo studies must have been conducted. 

Also, some corrections in language style are necessary, such as:

Line 166: However (and quite surprisingly) - either "Surprisingly" or, better yet, just "however"

Line 204: unfavorable adverse effects - adverse effects from the definition are unfavorable. 

Round 2

Reviewer 2 Report

I am happy with the replies.